# LEARNING DISCRETE REPRESENTATIONS TO UNDERSTAND AND PREDICT TISSUE BIOLOGY

## ABSTRACT

Learning tissue-level representations that capture the organization of entire tissues while preserving cellular and microenvironmental detail is a central challenge in spatial biology. While graph autoencoders have been employed to learn spatially aware continuous representations, they have limited utility for tissue-level generation, lack inherent interpretability for biological analysis, and are not readily reusable across contexts and modeling architectures. To address this challenge, we present **SQUINT**, a discrete representation learning framework for spatially-resolved transcriptomics that encodes tissues into a finite vocabulary of interpretable discrete codes. SQUINT achieves this by combining graph neural networks with vector quantization, conditioning on relative spatial distances, and employing a masking strategy during training. Cells are then represented by assignments to this shared vocabulary, allowing whole tissues to be modeled as sequences of discrete tokens. At inference, SQUINT codes enable cell imputation at arbitrary spatial locations outperforming state-of-the-art generative methods across diverse datasets. Further, we demonstrate the interpretability of these discrete tokens in capturing meaningful tissue structures beyond individual cells and reflecting recurrent microenvironmental organization patterns through downstream applications including 3D imputation, tumour stratification, and perturbation analysis.

## 1 INTRODUCTION

Spatially-resolved transcriptomics (SRT) has transformed spatial biology by enabling the measurement of gene expression at single-cell resolution while preserving tissue architecture (Karr et al., 2012; Zahedi et al., 2024; Du et al., 2023). Unlike single-cell RNA sequencing, which requires dissociation and thereby destroys spatial context (Yue et al., 2023), SRT captures both transcriptional states and their spatial locations, offering unprecedented opportunities to study how cells interact within microenvironments and how these interactions shape tissue function and disease.

A central challenge is learning tissue-level representations that capture organization across multiple scales, from cells to neighborhoods to whole tissue. Existing approaches such as SpaGCN (Hu et al., 2021), STAGATE (Dong & Yuan, 2022), and spaVAE (Tian et al., 2024) learn continuous embeddings from autoencoders for spatial domain discovery. Separately, transformer-based architectures have defined continuous tokens from histology images for imputation and labeling tasks (Wen et al., 2024b; Zhao et al., 2024; Long et al., 2023; Bao et al., 2025). While powerful, these approaches face three limitations: imaging-based data modalities rely on pixel-based transcriptional counts, continuous embeddings often obscure model semantics and are difficult to compare across tissues, and cell-level tokenizations lead to long sequences and limited interpretability at the tissue scale.

Discrete representation learning offers an appealing alternative. Vector quantization has shown that a finite codebook of codes can act as a compact, interpretable vocabulary in domains ranging from vision to graphs (van den Oord et al., 2017; Yang et al., 2023). We argue that SRT is an ideal setting for discretization because tissues are composed of recurring structural motifs such as layers, follicles, and crypts. Discrete codes provide symbolic units that describe both local microenvironments and global tissue organization. This design brings several benefits: (i) **Interpretability**, since discrete tokens act as human-readable identifiers of niches that can be traced across samples; (ii) **Stability**, since quantization avoids the rotational ambiguity of continuous embeddings; (iii) **Compactness**, by compressing noisy high-dimensional counts into denoised symbolic assignments; and (iv) **Com-**

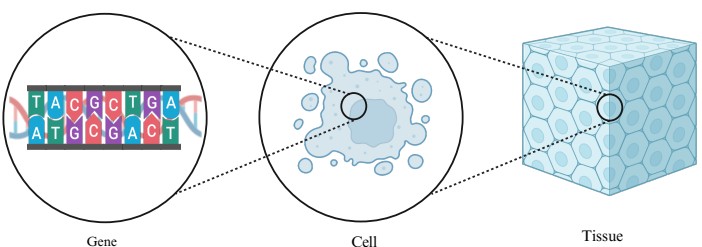

Figure 1: Tokenization across biological scales. Prior work represents genes with sequence tokens (e.g., nucleotide k-mers or regulatory peaks) and represents cells with gene tokens. SQUINT extends this hierarchy by representing tissues with spatially-aware cell tokens learned via a GNN, providing discrete and interpretable units for downstream generative modeling of tissues.

**positionality**, enabling tissues to be modeled as sequences of tokens that can serve as inputs to downstream generative or predictive models.

We introduce **SQUINT**, a spatially-aware discrete representation learning framework for SRT. SQUINT encodes tissues into a finite vocabulary of interpretable codes using a graph neural network encoder with vector quantization conditioned on relative spatial distances. Cells are represented by assignments to this shared vocabulary, allowing tissues to be modeled as sequences of discrete tokens (Figure 1). During training, we employ a masking strategy to encourage robust codes; at inference, these codes enable cell imputation at arbitrary spatial locations, effectively generating unseen microenvironments. We further show that the learned tokens capture meaningful tissue structures beyond individual cells, providing a symbolic representation of recurring microenvironments.

**Contributions.**

- SQUINT : a novel framework that couples graph neural networks with vector quantization for designing discrete tokens for cells based on high-resolution, single-cell transcriptomics data from MERFISH and Xenium assays across multiple organs and species.
- Discrete Cell Tokens: learnable shared vocabulary for compactness and stability across tissue slices for effective generation of entire gene expression profiles of unseen patches of cells, outperforming state-of-the-art methods.
- Interpretable Codebook: applications of the discrete tokens to showcase the inherent translational relevance for tumour stratification and in-silico cell-type perturbation analysis.

## 2 RELATED WORK

**Generative Cellular Genomics.** Data from scRNA-seq and FISH-based images modalities has been used to design continuous (Yang et al., 2022; Gong et al., 2023) and discrete (Li et al., 2025) tokens for gene expression reconstruction, clustering, etc., whereas our focus is on SRT data. Within SRT, scGPT-spatial (Wang et al., 2025) and CellPLM (Wen et al., 2024a) build discrete tokens for genes and cells respectively, for imputing expression of missing genes (among other tasks). However, they require cross-modal scRNA-seq data as reference for pre-training without which performance drops significantly (Wen et al., 2024a). Similarly, BLEEP (Xie et al., 2023), His2ST (Long et al., 2023), and ASIGN (Zhu et al., 2025) discretize tissues at region and spot levels based on paired SRT and reference histopathological 2D and 3D whole-slide images (WSI), respectively, for predicting gene expression. Wasserstein Flow Matching (Haviv et al., 2025) generates PCA representations of aggregate gene expression of tissue microenvironments, represented as distributions by conditioning on cell-type annotations using only SRT data. More recently, GeST (Hao et al., 2025) discretely quantizes cells from SRT data using PCA and K-means for use by transformers model for unseen cell generation.[1] Differently from these approaches, we build discrete tokens for cells within and across high-resolution, single-cell SRT tissue slices without query reference datasets from other

---

[1]We were unable to locate publicly available code for GeST.

modalities or cell-type annotations. We showcase their inherent generative capabilities by imputing expression profiles of entire unseen patches of cells and biological interpretability value through tumour stratification and cell-type perturbation analysis.

**Generative Graph Modeling.** Modern deep learning research into building graph generative models has broadly employed two paradigms, namely, (i) preserving adjacency using variational autoencoders (Simonovsky & Komodakis, 2018; Kipf & Welling, 2016), auto-regressive models (Li et al., 2018; You et al., 2018), and diffusion (Maron et al., 2019; Minello et al., 2024) for building small (up to 100 nodes) molecular graphs (Jin et al., 2018; Segler et al., 2018), (ii) pooling node-level representations from GNNs into graph representations and fusing them into input tokens for large-language models (Fatemi et al., 2024; Mao et al., 2024; Chen et al., 2024) for designing geometric conformations (Zhou et al., 2023) and generating scientific descriptions (Christofidellis et al., 2023), and Since SRT data is large ($> 10^5$ cells), high-dimensional ($> 10^3$ genes), and sparse (zero expression counts), and given the nature of our cross-tissue tasks such as tumour stratification and cell-level perturbation, the first two paradigms are less suitable. More recently, Ding et al. (2021) capture attributes and adjacency jointly via vector quantization leading to applications for molecule property prediction (Xia et al., 2023) and distillation (Yang et al., 2024). Our approach builds on this quantization paradigm by incorporating masking and conditioning on multiple covariates to generalize over previous approaches that focus on reconstruction, providing inherent interpretability with biological relevance.

## 3 PRELIMINARIES

**Spatial Transcriptomics and Graph Neural Networks.** Spatial transcriptomics data for a tissue is popularly modeled as a graph where each cell is a node, adjacency to other cells is determined by proximity in physical space, and gene expression counts are node attributes (Wang et al., 2021; Long et al., 2023). Let $[n]$ denote the set of integers from $1$ to $n$. Formally, we define a tissue $\mathcal{T}$ as an undirected, unweighted graph $\mathcal{G}_\mathcal{T} = (\mathcal{V}_\mathcal{T}, \mathcal{P}_\mathcal{T}, \mathbf{A}_\mathcal{T}, \mathbf{X}_\mathcal{T})$ comprising of a set of $n_\mathcal{T}$ cells $\mathcal{V}_\mathcal{T} = \{c_i\}_{i \in [n_\mathcal{T}]}$, their 2D spatial coordinates $\mathcal{P}_\mathcal{T} = \{(x_{c_i}, y_{c_i})\}_{i \in [n_\mathcal{T}]}$, adjacency matrix $\mathbf{A}_\mathcal{T} \in \{0, 1\}^{n_\mathcal{T} \times n_\mathcal{T}}$ connecting each cell to its $k$ nearest neighbors by Euclidean distance between their spatial coordinates (with additional reciprocal edges to ensure symmetry), and an attribute matrix $\mathbf{X}_\mathcal{T} \in \mathbb{Z}^{n_\mathcal{T} \times g_\mathcal{T}}$ of expression counts for $g_\mathcal{T}$ genes per cell[2]. Graph neural networks (GNNs) are widely used to learn node representations from such graph-structured data (Kipf, 2016; Hamilton et al., 2017; Veličković et al., 2017). The $l$-th layer of a message-passing GNN produces $d^{(l)}$-dimensional latent embeddings $\mathbf{H}^{(l)} \in \mathbb{R}^{n \times d^{(l)}}$ for cells by aggregating representations (e.g. mean) of their neighbours and combining them (e.g. concatenation) with their own representations from the previous layer. That is, for cell $c_i$:

$$\mathbf{h}_{c_i}^{(l)} = \text{COMBINE}\left(\mathbf{h}_{c_i}^{(l-1)}, \ \text{AGGREGATE}\left(\left\{\mathbf{h}_{c_j}^{(l-1)} : c_j \in \mathcal{N}(c_i)\right\}\right)\right) \tag{1}$$

where $\mathcal{N}(c_i)$ denotes the set of neighbors of $c_i$. Typically, $\mathbf{H}^{(0)} = \mathbf{X}$, and the final embeddings after $L$ layers, $\mathbf{H}^{(L)}$, are used for downstream tasks such as niche annotation (Birk et al., 2025).

**Graph Quantizers.** Vector Quantized Variational Autoencoders (VQ-VAEs) learn a mapping from input data onto a continuous distribution over a latent space followed by discretization to a categorical distribution over a set of codes in the same space such that the input data can be accurately reconstructed from these codes (Van Den Oord et al., 2017). Each node $c_i \in \mathcal{V}$ is first mapped to a latent embedding $\mathbf{h}_{c_i}^{(L)} \in \mathbb{R}^{d^{(L)}}$ via an $L$-layer GNN (Simonovsky & Komodakis, 2018). Denote a code as $\mathbf{e}_i \in \mathbb{R}^{d^{(L)}}$ and a set of $K$ such codes, collectively called a codebook, as $\mathbf{E} \in \mathbb{R}^{K \times d^{(L)}}$. Then, node embeddings are quantized to one of the $K$ code vectors using nearest-neighbor lookup:

$$\forall i \in [n], \ j_i^\star = \underset{j \in [K]}{\arg\min} \left\| \mathbf{h}_{c_i}^{(L)} - \mathbf{e}_j \right\|_2 \tag{2}$$

Denote $\mathbf{Z} \in \mathbb{R}^{n \times d^{(L)}}$ as the quantized embeddings where rows $\mathbf{z}_{c_i} = \mathbf{e}_{j_i^\star}, \forall i \in [n]$. Since $\arg\min$ is not differentiable, the stop-gradient sg operator (Bengio et al., 2013) is typically used to minimize

---

[2]To simplify notation, we omit the subscript $\mathcal{T}$, reinstating it only when multiple tissues are considered.

the distance between $\mathbf{H}^{(L)}$ and $\mathbf{Z}$ while preventing large fluctuations in code assignments:

$$\mathcal{L}_{\text{VQ}} = \underbrace{\frac{1}{n} \sum_{i=1}^{n} \left\| \text{sg} \left[ \mathbf{h}_{c_i}^{(L)} \right] - \mathbf{z}_{c_i} \right\|_2^2}_{\mathcal{L}_{\text{CODE}}} + \underbrace{\frac{1}{n} \sum_{i=1}^{n} \left\| \mathbf{h}_{c_i}^{(L)} - \text{sg} \left[ \mathbf{z}_{c_i} \right] \right\|_2^2}_{\mathcal{L}_{\text{COMMIT}}} \tag{3}$$

Further, $\mathbf{Z}$ is also jointly optimized to reconstruct attributes, $\hat{\mathbf{X}}$, using (say) mean scaled cosine error (Xia et al., 2023), adjacency, $\hat{\mathbf{A}}$, using entry-wise mean squared error Yang et al. (2024), predict labels (if available) using cross-entropy loss (Ding et al., 2021), or some combination thereof.

# 4  SQUINT

Our goal is to train informative codebooks for *generating* expression profiles of tissue microenvironments in unseen tissue regions. Section 4.1, defines the generative model, Section 4.2, describes the consequent architecture, and training and inference procedures.

## 4.1  GENERATIVE MODEL

**Masking.**  Since standard reconstruction criteria are unable to guarantee robust generation, denoised auto-encoders typically introduce point-wise zeros to mask the input (Vincent et al., 2010). With SRT being sparse, zero masking is not ideal since the model may learn to infer zero counts for all genes rather than identifying the masked site as the imputation target. Therefore, we introduce a custom masking strategy that (uniformly) randomly selects a subset of nodes denoted by $\mathbf{M} = (M_{c_1}, \ldots, M_{c_n}) \in \{0, 1\}^n$. The input attributes for these selected nodes are then replaced with a learnable mask token $\mathbf{m} \in \mathbb{R}^g$ ($\mathbf{A}$ is uncorrupted) plus a small noise term $\epsilon \sim \mathcal{N}\left(0, \sigma_\epsilon^2 I\right)$ as follows:

$$\tilde{\mathbf{x}}_{c_i} = (1 - M_{c_i}) \mathbf{x}_{c_i} + M_{c_i} (\mathbf{m} + \epsilon), \quad \tilde{\mathbf{X}} = \{\tilde{\mathbf{x}}_{c_i}\}_{i \in [n]} \tag{4}$$

**Inference Pathway.**  Denote $\mathbf{Z}$ as the latent variable (codes) and $\mathbf{C}$ as a set of covariates that encode semantic labels of interest such as cell-type annotations, relative spatial coordinates (e.g. RBF/Fourier on $\mathcal{P}$), among others. Consider the joint distribution $q_\phi\left(\mathbf{Z}, \tilde{\mathbf{X}}, \mathbf{X}, \mathbf{A}, \mathbf{C}\right)$ parameterized by $\phi$. Given tissue data $(\mathbf{X}, \mathbf{A}, \mathbf{C})$ from an empirical distribution, the inference pathway learns $\mathbf{Z}$ as follows:

$$\tilde{\mathbf{X}} \sim q_{\mathbf{m}}\left(\tilde{\mathbf{X}} \mid \mathbf{X}\right) \qquad \text{(masking)}$$

$$\mathbf{Z} \sim q_\phi\left(\mathbf{Z} \mid \tilde{\mathbf{X}}, \mathbf{A}, \mathbf{C}\right) \qquad \text{(learnable variational posterior)}$$

**Generation Pathway.**  Consider the joint distribution $p_\theta\left(\mathbf{X}_{\text{nbr}}, \mathbf{A}, \mathbf{Z}, \mathbf{C}\right)$ parameterized by $\theta$. Given observed covariates $\mathbf{C}$, the generation pathway generates data from the likelihood as follows:

$$\mathbf{Z} \sim p\left(\mathbf{Z} \mid \mathbf{C}\right) \qquad \text{(prior)}$$

$$\mathbf{A} \sim p_\theta\left(\mathbf{A} \mid \mathbf{Z}, \mathbf{C}\right) \qquad \text{(adjacency likelihood)}$$

$$\mathbf{X}_{\text{nbr}} \sim p_\theta\left(\mathbf{X}_{\text{nbr}} \mid \mathbf{A}, \mathbf{Z}, \mathbf{C}\right) \qquad \text{(attribute likelihood)}$$

In our experiments, we model $\mathbf{Z}$ to be independent of $\mathbf{C}$. Thus, $p\left(\mathbf{Z} \mid \mathbf{C}\right) = p\left(\mathbf{Z}\right)$. Further, we define the prior over $\mathbf{Z}$ as a uniform distribution over the discrete code indices and thus free of $\theta$. For future applications such as controllable generation for perturbations, this may be extended to a learned categorical prior (e.g. over spatial coordinates) (Maddison et al., 2017).

**Training Objective.**  Following Esmaeili et al. (2019), we frame the denoised, conditional lower bound (ELBO) as a minimization over the KL-divergence between the variational posterior and the variational prior. We model edge likelihood with a Bernoulli distribution. Since we are interested in microenvironments, we model $\mathbf{X}_{\text{nbr}}$, the total expression count of a cell and its neighbors for each gene, instead of $\mathbf{X}$ with the negative binomial distribution as in scVI (Gayoso et al., 2022). Putting this all together and taking the negative log-likelihood, we get the following objective for a single tissue:

$$\mathcal{L}_{\text{TOTAL}} = \lambda_{\mathbf{X}} \mathcal{L}_{\text{ATTR}}^{\text{NB}} + \lambda_{\mathbf{A}} \mathcal{L}_{\text{ADJ}}^{\text{BCE}} + \lambda_{\mathbf{E}} \mathcal{L}_{\text{VQ}} + \lambda_{\mathbf{m}} \|\mathbf{m}\|_2^2 \tag{5}$$

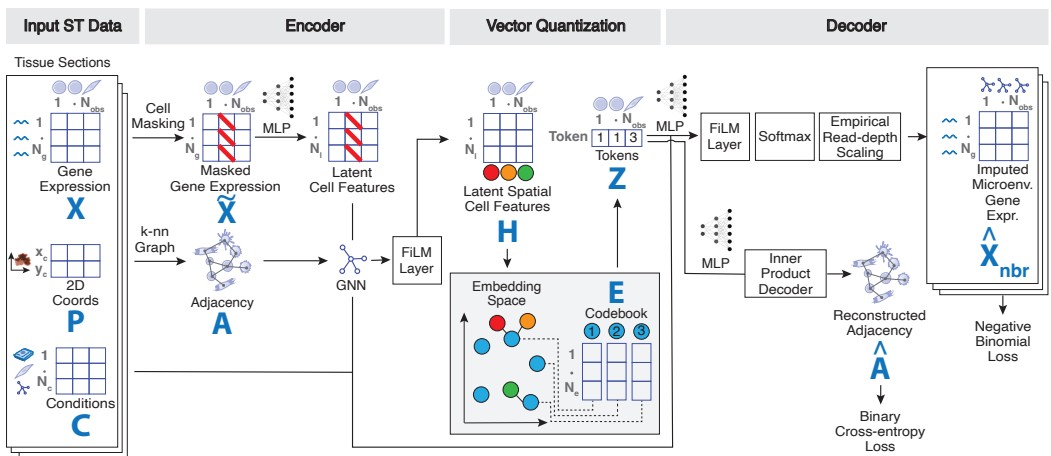

Figure 2: SQUINT comprises of an encoder, discretization bottleneck, and decoder to learn discrete cell tokens from SRT data. Given input tissue sections, gene expression vectors of cells are encoded to latent representations via an MLP (dimensionality reduction) followed by a GNN (neighborhood smoothing), followed by FiLM-style conditioning (e.g. using relative spatial distances and batch ID), to capture spatial dependencies and niche features. These are quantized via a codebook to yield one spatially-aware, discrete token per cell. The decoder conditions on tokens (via FiLM) to reconstruct (i) gene expression aggregated across the cell's microenvironment with empirical read-depth scaling and a negative binomial likelihood; in parallel, an inner-product decoder reconstructs the adjacency, trained with binary cross-entropy. The obtained tokens are discrete, interpretable representations of cellular state, contextualized by the cell's microenvironment, enabling spatial downstream analysis and generative modeling.

where the first two terms are negative binomial and binary cross entropy losses, respectively, the last term acts as regularization, and each $\lambda$ is the corresponding weight parameter. The training objective is the mean $\mathcal{L}_{\text{TOTAL}}$ over multiple tissues (see Appendix A.2 for the full derivation).

## 4.2 ARCHITECTURE

**Encoder.** The encoder captures the inference pathway to learn expressive codes. It parameterizes the variational posterior $q_\phi$ via a multi-layer perceptron (MLP) for dimensionality reduction on $\tilde{\mathbf{X}}$, followed by a GNN to learn neighborhood-aware latent representations for cells. In practice, we find that a 2-layer MLP helps handle sparsity and a single GraphSAGE (Hamilton et al., 2017) layer with neighbor sampling and mean pooling smoothes the noise in expression counts. Next, we define $\mathbf{C}$ using an RBF kernel to capture relative distances between neighbor cells because absolute coordinates of cells vary significantly across tissues depending on how they are assayed. We use feature-wise linear modulation (FiLM) (Perez et al., 2018) initialized with zero weights, bias and residual connections to condition the latent representations with $\mathbf{C}$. Lastly, we initialize a VQ module with a cosine codebook and straight-through estimator. The conditioned latents are quantized using K-means and the codebook is updated using exponential moving average (EMA) with a decay rate of 0.8. In our experiments, we find that a codebook size of 5000 with multiple heads improves codebook utilization and diversity and avoids collapse even in the absence of a supervised loss criterion.

**Decoder.** The generation pathway is captured by two decoders, one to reconstruct $\mathbf{A}$ from $\mathbf{Z}$ and $\mathbf{C}$, and another to generate microenvironment-aggregated gene expression $\mathbf{X}_{\text{nbr}}$ given $\mathbf{C}$, the learned codebook, and $\mathbf{A}$. Our adjacency decoder is an MLP that constructs edge vectors from the codebooks and computes their inner product followed by a sigmoid non-linearity to obtain edge probabilities. For reconstruction, $k$ highest probability edges are set to 1 and the rest to 0 with additional edges for ensuring undirectedness. Our attribute decoder is also an MLP that uses the assigned code and applies FiLM at each intermediate layer to condition the generation of $\mathbf{X}_{\text{nbr}}$ using $\mathbf{C}$. We apply softmax to the output to obtain a probability distribution over genes counts and scale it by the empirical read-depth

to obtain the final expression counts. This is helpful because variance in counts across cells and tissues can be large.

**Training and Inference.** In line with graph-learning literature, we train the encoder and decoders jointly in mini-batch fashion wherein a fraction of nodes are masked but the attribute loss is computed on all nodes. We observe robust results with a masking fraction of 0.2 annealed to 0.6 over the course of training. During inference, for each cell in the test patch, we impute its attributes by executing a forward pass of the model with the learned $\mathbf{m}$ in the presence of $\mathbf{C}$ to obtain a code assignment which is subsequently used to generate the counts.

## 5 EXPERIMENTS

In this section, we seek to quantitatively answer the following research questions:

1. **RQ1**: Is SQUINT capable of in-painting, i.e. generating expression of tissue microenvironments in completely unseen regions?
2. **RQ2**: Does SQUINT provide high quality tissue-level representations?
3. **RQ3**: Are the codes (tokens) generated by SQUINT biologically meaningful?

To this end, we evaluate SQUINT on four downstream tasks including 2D Imputation (Sec. 5.1), 3D Imputation (Sec. 5.2), Tumour Stratification (Sec. 5.3), and Cell-level Perturbation (Sec. 5.4). Our model is written in Pytorch and tested on 1 NVIDIA H100 64GB GPU card. Our code is available for public release as a Python `pip` package upon publication.

### 5.1 TASK A: 2D CELL IMPUTATION

Table 1: Summary statistics of all datasets.

|  | Cells | Edges | Genes | Sections | Cell Types | Assay |
|---|---|---|---|---|---|---|
| Brain (Mouse) | 48156 | 497936 | 1000 | 4 | 23 | MERFISH |
| Skin (Human) | 110278 | 1141520 | 1000 | 7 | 41 | Xenium |
| Kidney (Human) | 260193 | 2673293 | 1000 | 5 | NA | Xenium |

Table 2: Imputation quality of SQUINT with and without relative spatial distances as covariates compared to WFM (baseline) measured by Energy (E), Maximum Mean Discrepancy (MMD), Pearson Correlation (PC) metrics. Lower values for E and MMD and higher values for PC indicate better performance. Highlights in blue indicate best performance and NA indicates that the model is not applicable. Results are averaged across three random seeds.

| Model | Brain (Mouse) | | | Kidney (Human) | | | Skin (Human) | | |
|---|---|---|---|---|---|---|---|---|---|
| | E ↓ | MMD ↓ | PC ↑ | E ↓ | MMD ↓ | PC ↑ | E ↓ | MMD ↓ | PC ↑ |
| WFM | $0.1653 \pm 0.003$ | $0.1358 \pm 0.002$ | NA | NA | NA | NA | $0.1094 \pm 0.004$ | $0.0876 \pm 0.002$ | NA |
| SQUINT w/o C | $0.008 \pm 0.003$ | $0.045 \pm 0.004$ | $0.888 \pm 0.009$ | $0.004 \pm 0.000$ | $0.025 \pm 0.001$ | $0.827 \pm 0.016$ | $0.012 \pm 0.001$ | $0.086 \pm 0.005$ | $0.747 \pm 0.015$ |
| SQUINT | $0.005 \pm 0.001$ | $0.039 \pm 0.001$ | $0.916 \pm 0.008$ | $0.003 \pm 0.000$ | $0.019 \pm 0.001$ | $0.869 \pm 0.001$ | $0.013 \pm 0.001$ | $0.093 \pm 0.005$ | $0.788 \pm 0.010$ |

In this task, we evaluate the performance of SQUINT for imputing expression profiles of cellular microenvironments in unseen regions at user-specified spatial locations within tissue sections.

We conduct experiments on three datasets across two species, three organs, and two assays. BRAIN (MOUSE) has 4 sections of 1085 gene-atlas mouse brain totaling 48K cells assayed with MERFISH. SKIN 2D (HUMAN) includes 7 sections of skin tissue totaling 110K cells from 3 human patients assayed with a Xenium 5000 gene-panel. KIDNEY (HUMAN) covers 5 sections of kidney tissue totaling 260K cells from two human patients assayed with a Xenium 5000 gene-panel (cf. Table 1 for summary statistics).

In each case, we subset the data to top 1000 highly variable genes and define three patches within one section as imputation sites. SQUINT is trained on other cells from that section as well as the

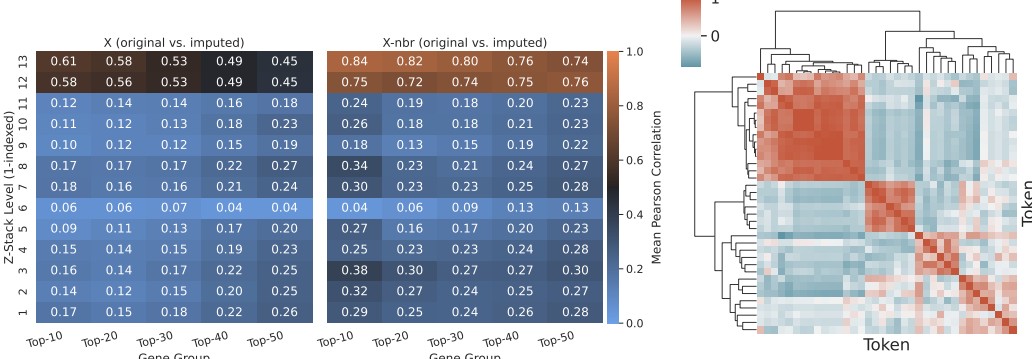

(a) Mean PC of top-k genes across Z-levels.   (b) Token co-occurrence across Z-levels.

Figure 3: Fig. 3a shows heatmaps of mean PC at each Z-level between original and imputed expression of globally ranked gene groups of the cell (left) and microenvironment (middle). Fig. 3b shows a PC clustermap between per-level token-usage profiles (proportions) for the top-50 most frequent tokens.

remaining sections (see App. B for a visual representation). We use batch IDs (unique for each section) and relative spatial distances obtained from an RBF kernel for conditioning the model during training and during inference. Further, we define a variant called SQUINT w/o $\mathbf{C}$ that does not use these conditions to measure their utility. We benchmark SQUINT against Wasserstein Flow Matching (WFM) Haviv et al. (2025) which is the only existing method for generating PCA representations of expression profiles for microenvironments.

We consider three metrics for evaluating the performance of SQUINT for imputing mean gene expression of tissue (1-hop) microenvironments. Pearson Correlation (PC) measures similarity point-wise at specific cell locations in the tissue. Mean Maximum Discrepancy (MMD) using an L1 Gaussian Total Variation kernel averaged across multiple bandwidths (Bunne et al., 2023) and the Energy Distance (E) (Heumos et al., 2024) to measure similarity in population distributions.

Table 2 showcases the imputation quality of WFM, SQUINT w/o $\mathbf{C}$, and SQUINT averaged across three random seeds. Since WFM is unsuited to imputing at particular spatial locations, PC is not applicable for WFM. Further, since cell-type labels are unavailable for the KIDNEY (HUMAN) dataset, results are unavailable for WFM which requires them as conditions during inference. SQUINT outperforms WFM and our variant without conditions on this task.

Further, we validate the design choices of the SQUINT architecture and training procedure through ablations on the backbone GNN, number of MLP layers in the SQUINT encoder, masking strategy, and imputation sites. Results are deferred to Appendix C.

## 5.2 TASK B: 3D SPATIAL IMPUTATION

SKIN 3D (HUMAN) is a collection of 13 sections of skin tissue totaling nearly 280K cells from a single human patient assayed with a Xenium 5000 gene-panel. These 13 sections are approximately $60\mu$ apart along the vertical (Z-level) axis and may be viewed jointly as a 3D model of the tissue. We trained a SQUINT model on this dataset using a codebook size of 50 and 200 latent dimensions across 13 sections wherein 10% cells at each Z-level were held out for evaluation.

For Fig. 3a, we computed Pearson correlation per gene between the original and imputed expression cell-wise (left) and microenvironment-wise (middle) using all cells. Genes were ranked by this global correlation and grouped into top-$k$ sets. For each Z-level, we computed gene-wise correlations using held-out test cells from that level and reported the mean across genes in the group. Across levels, imputation quality tends to be higher for $\mathbf{X}_{\text{nbr}}$ than for $\mathbf{X}$, indicating that neighborhood-aggregated structure is better captured by the model than per-cell signals. Within each panel, mean correlation decreases as the size of the gene group increases, reflecting the rising difficulty in imputation quality. Further, we note that imputation quality deteriorates towards the middle of the Z-stack and improves

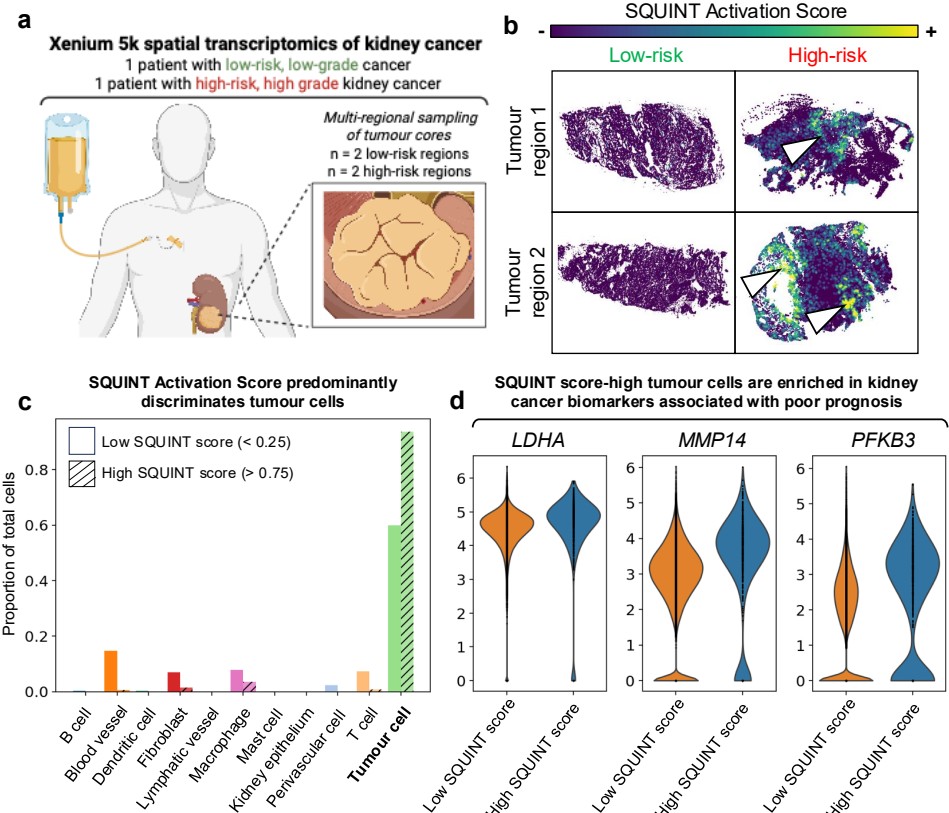

Figure 4: Kidney Tumour Stratification. (a) SQUINT was trained on 260,153 cells from five tissue sections across two patients with kidney cancer. (b) SQUINT activation scores show high activation in tumour cores of the high-risk patient but not the low-risk case. (c) High-score cells (>0.75) are enriched for tumour epithelial populations compared to low-score cells (<0.25). (d) Within tumour epithelial cells, high-score cells show differential expression of genes linked to aggressive kidney cancer (LDHA, MMP14, PFKFB3).

towards the higher levels. This may be due to the distance between the sections being larger towards the middle of the Z-stack or due to mis-alignment of the sections.

In Fig. 3b, each block captures correlation between token pairs based on their co-occurrence across Z-levels. Values close to 1 indicate token pairs likely to be enriched at the same levels while values closer to -1 indicate anti-co-occurrence across levels. Dendrograms group tokens into co-occurring "communities". The token co-occurrence map reveals coherent groups of tokens whose usage co-varies with depth suggesting shared spatial or biological roles captured by the discrete codes.

## 5.3 TASK C: KIDNEY TUMOUR STRATIFICATION

In this task, our goal is determine if codes generated by SQUINT have translational relevance for tumor stratification. We trained SQUINT on all 5 sections of two patients in KIDNEY (HUMAN) and conditioned on relative spatial coordinates of cells along with the batch IDs (Figure 5a). Clinical annotations and cell-type annotations were withheld from model training (Li et al., 2022b). One patient had a tumour classified as low clinical risk of developing metastasis, based on histological and morphological assessment, while the second patient presented with high-risk disease and metastasis. At inference, we identified the top 5 codes with the most frequent assignment among cells in the high-risk patient. Next, we computed activation scores for each cell across both patients as the number of times the cell or its 1-hop microenvironment was assigned to the top 5 codes normalized by the number of neighboring cells.

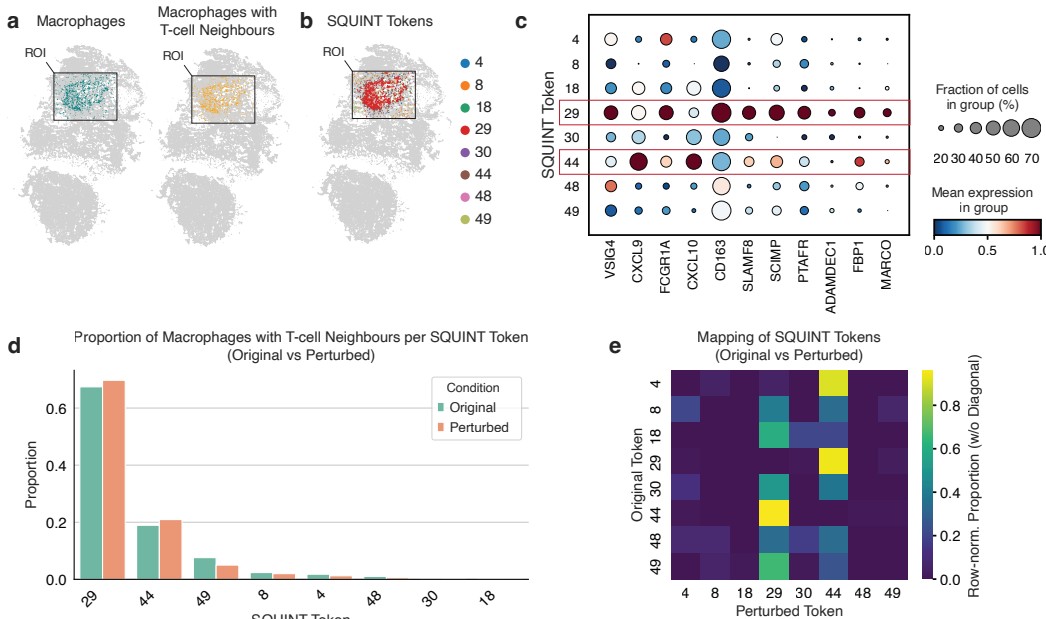

Figure 5: SQUINT tokens recapitulate relevant macrophage biology and cell state changes after perturbation of neighbouring T-cells. (a) Kidney tissue section with marked tertiary lymphoid structure region of interest (ROI), displaying macrophages and the subset with T-cell neighbours. (b) SQUINT token assignments for macrophages in the ROI. (c) Token-level expression of a macrophage gene program comprising genes relevant in the context of immune checkpoint blockade (ICB): dot size shows the fraction of macrophages in each token expressing a marker, and color shows mean expression. (d) Bar plot displaying proportions of macrophages with T-cell neighbours before and after a perturbation mirroring ICB. After perturbation, macrophages transition to tokens 29 and 44, characterized by high expression in the post-ICB signature. (e) Row-normalized proportions (excluding diagonal) of original and perturbed token assignments highlight transitions.

This revealed distinct spatial patterns: regions of high activation were confined to tumour cores in the high-risk patient, whereas no such enrichment was observed in the low-risk tumour (Figure 5b). To examine the cellular context of these signals, we compared cell type proportions between cells with low ($< 0.25$) and high ($> 0.75$) activation scores. High-score cells were markedly enriched for tumour epithelial populations (Figure 5c). Restricting the analysis to tumour epithelial cells, we identified differentially expressed genes (DEGs) between high- and low-score groups. The high-score compartment was enriched for DEGs previously linked to aggressive kidney cancer biology and adverse outcomes, including LDHA (Girgis et al., 2014), MMP14 (Zhao et al., 2022) and PFKB3 (Li et al., 2022a) (Figure 5d). Together, these results highlight the ability of SQUINT to achieve clinically meaningful tumour stratification directly from spatial transcriptomic profiles.

## 5.4 TASK D: CELL-LEVEL PERTURBATION

In this task, we trained a separate SQUINT model on the 5 KIDNEY sections introduced previously and ran inference before and after a cell-level perturbation to investigate the biological relevance of the discrete tokens. Specifically, we computationally mirrored immune checkpoint blockade (ICB) treatment by replacing the gene expression vectors of native T-cells in a tertiary lymphoid structure region of interest (ROI) with a prototype gene expression vector of activated T-cells after ICB treatment. We used the annotations from (Akbarnejad et al., 2025) who have identified native T-cells and some post-ICB-like activated T-cells in this dataset. We then computed the mean gene expression vector across all post-ICB-like activated T-cells and used it to replace the gene expression vectors of all native T-cells in the TLS region. We used the trained SQUINT encoder to obtain tokens before and after perturbation and compared the tokens of macrophages in the neighbourhood of T-cells

as these are expected to react to the induced T-cell changes by transitioning to a known post-ICB gene signature (Akbarnejad et al., 2025).

First, obtaining the tokens of all macrophages in the ROI in the unperturbed scenario, we observed that specific SQUINT codes (codes 29 and 44) represented high expression in the curated gene signature, highlighting how our discrete tokens capture relevant biology and offer inherent interpretability without time-consuming downstream analysis. After the perturbation, macrophages with T-cells in their neighborhood, i.e. macrophages affected by the perturbation, moved towards these tokens representing high expression in the post-ICB gene signature.

## 6 CONCLUSIONS AND FUTURE WORK

Existing approaches for tokenizing genes for SRT data can be inadequate for tissues-level tasks. As a remedy, we propose a novel approach called SQUINT to model cells as discrete tokens (codes) using a graph-based vector-quantized variational autoencoder. We showcase the generative power of these trained codebooks for imputing expression profiles of entire unseen patches of cells used a learned mask and conditioned on covariates such relative spatial positions to achieve superior performance over a state-of-the-art method. Additionally, we showcase the translational relevance of these codes on a variety of downstream tasks ranging from 3D imputation of Z-stack tissue sections, tumour stratification in kidney patients, and cell-type gene profile perturbations. Thus, we argue for their use stand-alone or in conjunction with tokens from other modalities for training downstream transformer models for generative applications.

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

## A  GENERATIVE MODEL

### A.1  FACTORIZATION OF THE JOINT DISTRIBUTIONS

In this section discuss the factorized joint distributions for the inference and generation pathways. These distributions are subsequently used for defining the training objective in Section **??**. The joint likelihood for the variational posterior for the inference pathway is defined over model parameters $\phi$ and $\mathbf{m}$. By the chain rule for conditional probability, we have:

$$q_{\phi,\mathbf{m}}\left(\mathbf{Z},\tilde{\mathbf{X}},\mathbf{X},\mathbf{A},\mathbf{C}\right) = q_{\phi,\mathbf{m}}\left(\mathbf{Z}\mid\tilde{\mathbf{X}},\mathbf{X},\mathbf{A},\mathbf{C}\right)q_{\phi,\mathbf{m}}\left(\tilde{\mathbf{X}}\mid\mathbf{X},\mathbf{A},\mathbf{C}\right)q_{\phi,\mathbf{m}}\left(\mathbf{X},\mathbf{A},\mathbf{C}\right)$$

Note, $\mathbf{X}$, $\mathbf{A}$, and $\mathbf{C}$ are independent of $\phi$ and $\mathbf{m}$. For notational convenience, we drop the parameters from the subscript, i.e. $q_{\phi,\mathbf{m}}\left(\mathbf{X},\mathbf{A},\mathbf{C}\right) = q\left(\mathbf{X},\mathbf{A},\mathbf{C}\right)$.

From Equation 4, the corrupted attributes $\tilde{\mathbf{X}}$ depends only on $\mathbf{X}$ and $\mathbf{M}$, and are independent of $\mathbf{A}$ and $\mathbf{C}$. $\tilde{\mathbf{X}}$ is also independent of $\phi$. Therefore:

$$q_{\phi,\mathbf{m}}\left(\tilde{\mathbf{X}}\mid\mathbf{X},\mathbf{A},\mathbf{C}\right) = q_{\mathbf{m}}\left(\tilde{\mathbf{X}}\mid\mathbf{X}\right)$$

Let $T$ denote the training epoch and $B_T = \sum_i M_{c_i}$ denote the masking budget for epoch $T$. The mask index vector $\mathbf{M}$ is independently drawn from $q\left(\mathbf{M}\mid T\right)$ over all $B_T$-sized subsets of $n$, independent of $\mathbf{X}$, $\mathbf{A}$, and $\mathbf{C}$. That is,

$$\mathbf{M}\perp\left(\mathbf{X},\mathbf{A},\mathbf{C}\right)\mid T$$

Since the $\mathbf{Z}$ does not observe $\mathbf{M}$ or $\mathbf{X}$ directly, it can be marginalized out. That is,

$$\mathbf{Z}\perp\left(\tilde{\mathbf{X}},\mathbf{A},\mathbf{C}\right)\mid\mathbf{X},\mathbf{M}$$

Putting this all together, we can write the factorized joint distribution for the inference pathway as:

$$q_{\phi,\mathbf{m}}\left(\mathbf{Z},\tilde{\mathbf{X}},\mathbf{X},\mathbf{A},\mathbf{C}\right) = q_{\phi}\left(\mathbf{Z}\mid\tilde{\mathbf{X}},\mathbf{A},\mathbf{C}\right)q_{\mathbf{m}}\left(\tilde{\mathbf{X}}\mid\mathbf{X}\right)q\left(\mathbf{X},\mathbf{A},\mathbf{C}\right) \quad (6)$$

The factorization of the joint likelihood for the variational prior over parameters $\theta$ for the generation pathway follows directly from the chain rule for conditional probability:

$$p_{\theta}\left(\mathbf{X},\mathbf{A},\mathbf{Z},\mathbf{C}\right) = p_{\theta}\left(\mathbf{X}\mid\mathbf{A},\mathbf{Z},\mathbf{C}\right)p_{\theta}\left(\mathbf{A}\mid\mathbf{Z},\mathbf{C}\right)p_{\theta}\left(\mathbf{Z}\mid\mathbf{C}\right)p\left(\mathbf{C}\right) \quad (7)$$

We drop the subscript for $p_{\theta}\left(\mathbf{C}\right)$ since $\mathbf{C}$ is independent of $\theta$.

### A.2  DERIVATION OF THE DENOISED, CONDITIONAL ELBO

Following Esmaeili et al. (2019), we derive the denoised, conditional ELBO by minimizing the KL-divergence between the variational posterior and the variational prior.

$$\max_{\theta,\phi,\mathbf{m}} -\mathcal{D}_{\mathrm{KL}}\left[q_{\phi,\mathbf{m}}\left(\mathbf{Z},\tilde{\mathbf{X}},\mathbf{X},\mathbf{A},\mathbf{C}\right)\,\middle\|\,p_{\theta}\left(\mathbf{X},\mathbf{A},\mathbf{Z},\mathbf{C}\right)\right]$$

$$\max_{\theta,\phi,\mathbf{m}} -\int_{\left(\mathbf{Z},\tilde{\mathbf{X}},\mathbf{X},\mathbf{A},\mathbf{C}\right)} q_{\phi,\mathbf{m}}\left(\mathbf{Z},\tilde{\mathbf{X}},\mathbf{X},\mathbf{A},\mathbf{C}\right)\log\frac{q_{\phi,\mathbf{m}}\left(\mathbf{Z},\tilde{\mathbf{X}},\mathbf{X},\mathbf{A},\mathbf{C}\right)}{p_{\theta}\left(\mathbf{X},\mathbf{A},\mathbf{Z},\mathbf{C}\right)}d\left(\mathbf{Z},\tilde{\mathbf{X}},\mathbf{X},\mathbf{A},\mathbf{C}\right)$$

$$\max_{\theta,\phi,\mathbf{m}} \underbrace{\int_{\left(\mathbf{Z},\tilde{\mathbf{X}},\mathbf{X},\mathbf{A},\mathbf{C}\right)} q_{\phi,\mathbf{m}}\left(\mathbf{Z},\tilde{\mathbf{X}},\mathbf{X},\mathbf{A},\mathbf{C}\right)\log\frac{p_{\theta}\left(\mathbf{X},\mathbf{A},\mathbf{Z},\mathbf{C}\right)}{q_{\phi,\mathbf{m}}\left(\mathbf{Z},\tilde{\mathbf{X}},\mathbf{X},\mathbf{A},\mathbf{C}\right)}d\left(\mathbf{Z},\tilde{\mathbf{X}},\mathbf{X},\mathbf{A},\mathbf{C}\right)}_{\mathcal{J}(\theta,\phi,\mathbf{m})} \quad (8)$$

Writing this joint distribution as a nested integral based on the factorized joint distributions for the inference and generation pathways from Equation 6 and Equation 7, respectively, we have:

$$\mathcal{J}\left(\theta,\phi,\mathbf{m}\right) = \int q\left(\mathbf{X},\mathbf{A},\mathbf{C}\right)\Bigg\{\int q_{\mathbf{m}}\left(\tilde{\mathbf{X}}\mid\mathbf{X}\right)\bigg[\int q_{\phi}\left(\mathbf{Z}\mid\tilde{\mathbf{X}},\mathbf{A},\mathbf{C}\right)\Big(\log p_{\theta}\left(\mathbf{X}\mid\mathbf{A},\mathbf{Z},\mathbf{C}\right)$$

$$+\log p_{\theta}(\mathbf{A}\mid\mathbf{Z},\mathbf{C}) + \log p_{\theta}(\mathbf{Z}\mid\mathbf{C}) + \log p\left(\mathbf{C}\right) - \log q_{\phi}\left(\mathbf{Z}\mid\tilde{\mathbf{X}},\mathbf{A},\mathbf{C}\right)$$

$$-\log q_{\mathbf{m}}\left(\tilde{\mathbf{X}}\mid\mathbf{X}\right) - \log q\left(\mathbf{X},\mathbf{A},\mathbf{C}\right)\Big)d\mathbf{Z}\bigg]d\tilde{\mathbf{X}}\Bigg\}d\left(\mathbf{X},\mathbf{A},\mathbf{C}\right) \quad (9)$$

$\log \mathrm{p}\left(\mathbf{C}\right)$ and $\log \mathrm{q}\left(\mathbf{X}, \mathbf{A}, \mathbf{C}\right)$ are independent of $\theta$, $\mathbf{m}$, and $\phi$, and so integrating them out yields constants with respect to the optimization. Rearranging the terms, we get:

$$
\begin{aligned}
\mathcal{J}\left(\theta, \phi, \mathbf{m}\right) = \int \mathrm{q}\left(\mathbf{X}, \mathbf{A}, \mathbf{C}\right) &\bigg\{ \int \mathrm{q_m}\left(\tilde{\mathbf{X}} \mid \mathbf{X}\right) \bigg[ \int \mathrm{q}_\phi\left(\mathbf{Z} \mid \tilde{\mathbf{X}}, \mathbf{A}, \mathbf{C}\right)\left( -\log \mathrm{q_m}\left(\tilde{\mathbf{X}} \mid \mathbf{X}\right) \right. \\
&+ \log \mathrm{p}_\theta\left(\mathbf{X} \mid \mathbf{A}, \mathbf{Z}, \mathbf{C}\right) + \log \mathrm{p}_\theta(\mathbf{A} \mid \mathbf{Z}, \mathbf{C}) \\
&+ \left. \log \mathrm{p}_\theta(\mathbf{Z} \mid \mathbf{C}) - \log \mathrm{q}_\phi\left(\mathbf{Z} \mid \tilde{\mathbf{X}}, \mathbf{A}, \mathbf{C}\right) \right)\mathrm{d}\mathbf{Z} \bigg]\mathrm{d}\tilde{\mathbf{X}} \bigg\} \mathrm{d}\left(\mathbf{X}, \mathbf{A}, \mathbf{C}\right) \\
&+ \mathrm{const.}
\end{aligned}
\tag{10}
$$

Since, $\log \mathrm{q_m}\left(\tilde{\mathbf{X}} \mid \mathbf{X}\right)$ is independent of $\mathbf{Z}$:,

$$
\int \mathrm{q}_\phi\left(\mathbf{Z} \mid \tilde{\mathbf{X}}, \mathbf{A}, \mathbf{C}\right)\left( -\log \mathrm{q_m}\left(\tilde{\mathbf{X}} \mid \mathbf{X}\right) \right)\mathrm{d}\mathbf{Z} = -\log \mathrm{q_m}\left(\tilde{\mathbf{X}} \mid \mathbf{X}\right) \underbrace{\left( \int \mathrm{q}_\phi\left(\mathbf{Z} \mid \tilde{\mathbf{X}}, \mathbf{A}, \mathbf{C}\right)\mathrm{d}\mathbf{Z} \right)}_{1}
$$

$$
= -\log \mathrm{q_m}\left(\tilde{\mathbf{X}} \mid \mathbf{X}\right)
\tag{11}
$$

Further:

$$
\int \mathrm{q}_\phi\left(\mathbf{Z} \mid \tilde{\mathbf{X}}, \mathbf{A}, \mathbf{C}\right)\left[ \log \mathrm{p}_\theta(\mathbf{Z} \mid \mathbf{C}) - \log \mathrm{q}_\phi\left(\mathbf{Z} \mid \tilde{\mathbf{X}}, \mathbf{A}, \mathbf{C}\right) \right] \mathrm{d}\mathbf{Z}
$$

$$
= -\mathcal{D}_{\mathrm{KL}}\left[ \mathrm{q}_\phi\left(\mathbf{Z} \mid \tilde{\mathbf{X}}, \mathbf{A}, \mathbf{C}\right) \,\Big\|\, \mathrm{p}_\theta(\mathbf{Z} \mid \mathbf{C}) \right]
\tag{12}
$$

Plugging in Equation 11 and Equation 12 into Equation 9, we get:

$$
\begin{aligned}
\mathcal{J}\left(\theta, \phi, \mathbf{m}\right) = &-\int \mathrm{q}\left(\mathbf{X}, \mathbf{A}, \mathbf{C}\right)\bigg\{ \int \mathrm{q_m}\left(\tilde{\mathbf{X}} \mid \mathbf{X}\right)\left( \log \mathrm{q_m}\left(\tilde{\mathbf{X}} \mid \mathbf{X}\right) \right)\mathrm{d}\tilde{\mathbf{X}} \bigg\} \mathrm{d}\left(\mathbf{X}, \mathbf{A}, \mathbf{C}\right) \\
&+ \int \mathrm{q}\left(\mathbf{X}, \mathbf{A}, \mathbf{C}\right)\bigg\{ \int \mathrm{q_m}\left(\tilde{\mathbf{X}} \mid \mathbf{X}\right)\bigg[ \left( \int \mathrm{q}_\phi\left(\mathbf{Z} \mid \tilde{\mathbf{X}}, \mathbf{A}, \mathbf{C}\right)\left( \log \mathrm{p}_\theta\left(\mathbf{X} \mid \mathbf{A}, \mathbf{Z}, \mathbf{C}\right) \right.\right. \\
&\qquad\qquad \left.\left. + \log \mathrm{p}_\theta(\mathbf{A} \mid \mathbf{Z}, \mathbf{C}) \right)\mathrm{d}\mathbf{Z} \right)\bigg]\mathrm{d}\tilde{\mathbf{X}} \bigg\} \mathrm{d}\left(\mathbf{X}, \mathbf{A}, \mathbf{C}\right) \\
&- \int \mathrm{q}\left(\mathbf{X}, \mathbf{A}, \mathbf{C}\right)\bigg\{ \int \mathrm{q_m}\left(\tilde{\mathbf{X}} \mid \mathbf{X}\right)\left( \mathcal{D}_{\mathrm{KL}}\left[ \mathrm{q}_\phi\left(\mathbf{Z} \mid \tilde{\mathbf{X}}, \mathbf{A}, \mathbf{C}\right) \,\Big\|\, \mathrm{p}_\theta(\mathbf{Z} \mid \mathbf{C}) \right] \right)\mathrm{d}\tilde{\mathbf{X}} \bigg\} \mathrm{d}\left(\mathbf{X}, \mathbf{A}, \mathbf{C}\right) \\
&+ \mathrm{const.}
\end{aligned}
\tag{13}
$$

Using the definition of conditional expectation, we obtain the training objective:

$$
\begin{aligned}
&\underset{\theta, \phi, \mathbf{m}}{\arg \min} \quad -\mathcal{J}\left(\theta, \phi, \mathbf{m}\right) \\
&\underset{\theta, \phi, \mathbf{m}}{\arg \min} \quad - \underbrace{\mathbb{E}_{\mathrm{q}(\mathbf{X}, \mathbf{A}, \mathbf{C})}\mathbb{E}_{\mathrm{q_m}(\tilde{\mathbf{X}}|\mathbf{X})}\mathbb{E}_{\mathrm{q}_\phi(\mathbf{Z}|\tilde{\mathbf{X}}, \mathbf{A}, \mathbf{C})}\left[ \log \mathrm{p}_\theta\left(\mathbf{X} \mid \mathbf{A}, \mathbf{Z}, \mathbf{C}\right) + \log \mathrm{p}_\theta(\mathbf{A} \mid \mathbf{Z}, \mathbf{C}) \right]}_{\text{reconstruction error}} \\
&\qquad\qquad + \underbrace{\mathbb{E}_{\mathrm{q}(\mathbf{X}, \mathbf{A}, \mathbf{C})}\mathbb{E}_{\mathrm{q_m}(\tilde{\mathbf{X}}|\mathbf{X})}\left[ \mathcal{D}_{\mathrm{KL}}\left[ \mathrm{q}_\phi\left(\mathbf{Z} \mid \tilde{\mathbf{X}}, \mathbf{A}, \mathbf{C}\right) \,\Big\|\, \mathrm{p}_\theta(\mathbf{Z} \mid \mathbf{C}) \right] \right]}_{\text{latent entropy}} \\
&\qquad\qquad + \underbrace{\mathbb{E}_{\mathrm{q}(\mathbf{X}, \mathbf{A}, \mathbf{C})}\mathbb{E}_{\mathrm{q_m}(\tilde{\mathbf{X}}|\mathbf{X})}\left[ \log \mathrm{q_m}\left(\tilde{\mathbf{X}} \mid \mathbf{X}\right) \right]}_{\text{corruption variance}}
\end{aligned}
\tag{14}
$$

## B    EXPERIMENT SETTINGS

### B.1    DATASETS.

For our results in Table 2, we define three patches within one section as imputation sites. Figure **??** shows the train-test splits for the Brain (Mouse) dataset and Figure **??** shows the train-test splits for the Kidney (Human) dataset.

### B.2    OVERSMOOTHING.

In our experiments, we set the number of MLP layers in the SQUINT encoder to 2 and the number of GNN layers to 1. This does not lead to oversmoothing because the GNN layer is able to capture the neighborhood-aware information without homogenizing cell microenvironment representations. This is consistent with the findings of Alon & Yahav (2021) that the bottleneck of GNNs since the diameter and average degree of the spatial neighborhood graphs built from 8-nearest neighbors is larger ($> 50$ and $> 20$, respectively) than the depth of the convolutional layers. We further find that the performance of the model degrades beyond 3 GNN layers, while 2 GNN layers offers marginal improvement in imputation quality over 1 GNN layer at the cost of increased computational complexity.

### B.3    USAGE.

We provide peak memory usage and runtime required for training SQUINT for the different datasets in Table 3.

Table 3: Peak memory usage and runtime required for training SQUINT for the different datasets.

| Dataset | Peak Memory | Runtime |
|---|---|---|
| Brain (Mouse) | 4.17 GB | 2 minutes 52 seconds |
| Kidney (Human) | 15.83 GB | 16 minutes 53 seconds |
| Skin (Human) | 16.46 GB | 6 minutes 1 seconds |

## C    ABLATIONS

In this section, we provide four additional ablations across two tasks and two datasets to evaluate the performance of our proposed method. For each ablation, we change one component of SQUINT, execute the train-test pipeline across three random seeds, and measure the impact of the change on the performance of the model.

### C.1    BACKBONE GNN

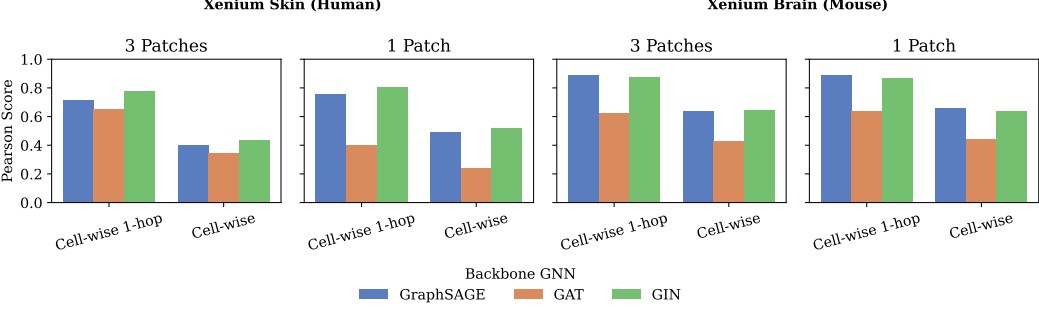

Figure 6: Comparison of the performance of the different backbone GNNs on the Xenium Skin (Human) and Xenium Brain (Mouse) datasets.

We replace GRAPHSAGE with GAT and GIN as the backbone GNN module within the SQUINT encoder to compute the latent representations of the nodes. We retrained the model for 20 epochs and computed the Pearson-correlation scores between the original and imputed attributes for 1 and 3 test patches each for the Xenium Skin (Human) and Xenium Brain (Mouse) datasets. Figure 6 shows the performance of the different backbone GNNs on the Xenium Skin (Human) and Xenium Brain (Mouse) datasets. GRAPHSAGE and GIN are consistently similar in performance across metrics, number of test patches, and metrics across both datasets with GIN slightly better than GRAPHSAGE on Xenium Skin (Human) and GRAPHSAGE slightly better than GIN on Xenium Brain (Mouse). In all cases, GAT shows significantly lower performance compared to GRAPHSAGE and GIN.

## C.2 NUMBER OF MLP LAYERS

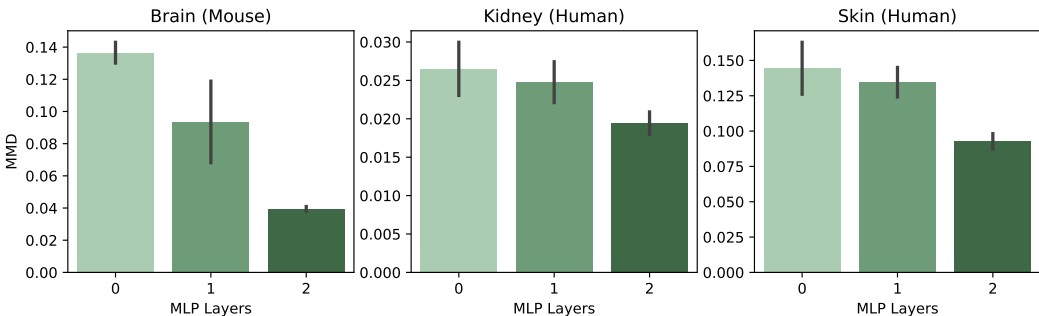

Figure 7: Impact of number of MLP layers in the SQUINT encoder on imputation quality measured by mean MMD scores averaged across 3 random seeds (lower values indicate better performance).

In Table 2, we reported results for an encoder with 2 MLP layers. In Figure 7 we build an encoder with no MLP layers (only one GNN layer) and 1 MLP layer to evaluate the impact of the number of MLP layers on the imputation quality. Results show that an encoder with 2 MLP layers offers better performance than 0 and 1 layers.

## C.3 MASKING STRATEGY

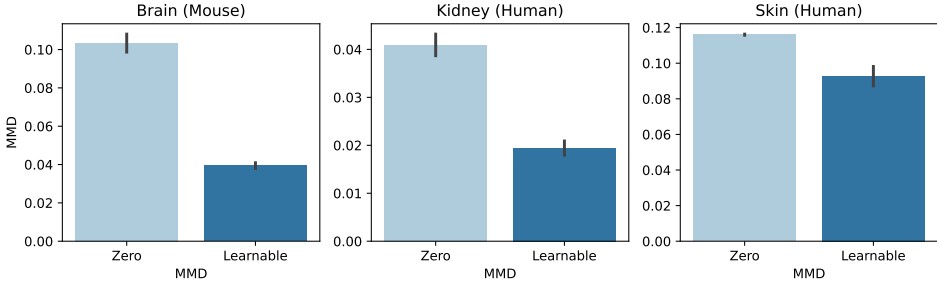

Figure 8: Impact of the masking strategy on imputation quality measured by mean MMD scores averaged across 3 random seeds (lower values indicate better performance).

As we discussed in Section 4.1, during training, we replace attributes of selected nodes with a learnable mask token plus a small noise term. Then, during inference, we generate expression profiles for cellular microenvironments at the imputation target sites by executing a forward pass of the model with the learned mask token in the presence of $\mathbf{C}$. In Figure 8, we conduct an ablation study on the masking strategy by using an all-zero mask instead of a learnable mask. Results show that using a learnable mask token leads to better imputation quality (MMD scores).

Further, we note our datasets do not contain any zero-expression cells. However, using a zero-mask causes the model to infer zero expression for 7.2%, 3.1%, and 9.9% of cells from the imputation patches of the BRAIN (MOUSE), KIDNEY (HUMAN), and SKIN 2D (HUMAN) datasets, respectively.

This represents a crucial limitation of the zero-mask strategy. In these cases, we define $\lambda_{\text{zero}}$ as a penalty term computed empirically as the mean maximum discrepancy distance between the ground-truth expression distribution and a canonical uninformed uniform distribution and the overall MMD score is computed as the sum of the non-zero expression imputed cells and the fraction of zero-expression imputed cells multiplied by $\lambda_{\text{zero}}$.

### C.4    IMPUTATION SITES

Table 4: Impact of the choice of imputation sites, random cells and contiguous patches, on imputation quality (MMD scores, lower values are better).

|  | Brain (Mouse) | Kidney (Human) | Skin (Human) |
|---|---|---|---|
| Random | $0.014 \pm 0.001$ | $0.008 \pm 0.000$ | $0.013 \pm 0.000$ |
| 3 Patches | $0.039 \pm 0.001$ | $0.019 \pm 0.001$ | $0.093 \pm 0.005$ |

In Table 4, we report MMD scores (lower values indicate better performance) for imputing for random cells evenly distributed across tissue sections and 3 contiguous patches of cells (i.e. the setting in Table 2). The large performance gap between the two settings indicates the lower difficulty of the random imputation setting since the model is able to access the neighborhood around each imputation site compared to when entire neighborhoods of cells within patches are masked.

## D    AI USAGE CLARIFICATION.

We used LLM agents such as Cursor for assisting in code development and ChatGPT for polishing text for improved readability and grammar. We take responsibility for suggestions from AI models that we incorporated into our work. All methodological contributions come from authors.

