# OpenReview forum: "LEARNING DISCRETE REPRESENTATIONS TO UNDER- STAND AND PREDICT TISSUE BIOLOGY"
_ICLR.cc/2026/Conference — Submitted to ICLR 2026_

### Official Review · Reviewer_yXRW · 2025-10-18

**Soundness:** 1
**Presentation:** 2
**Contribution:** 2
**Rating:** 2
**Confidence:** 5

**Summary:**

The paper proposes SQUINT, a discrete representation learning framework for spatially resolved transcriptomics (SRT). The method combines a GNN encoder, vector quantization, and conditional masking to tokenize cells into discrete codes. The learned tokens are used for tissue-level generative modeling, tumor stratification, and perturbation analysis.

**Strengths:**

- The paper is easy to follow.
- The idea of learning discrete spatially-aware cell tokens for SRT is conceptually interesting.

**Weaknesses:**

- The evaluation of SQUINT is weak, which can be summarized in several aspects:
  - For Task A (2D imputation), the author includes the Wasserstein Flow Matching (WFM) as a baseline for comparison. However,  it is apparent that WFM is not a suitable baseline as there are too many "NA" in the table. I suggest that the authors try simpler baselines such as Gaussian-process-based spatial interpolation methods, which have been demonstrated to perform effectively for spatial imputation in [2–4].
  - For Task B-D, quantitative comparison with baseline methods seems to be completely missing.
  - The ablation studies for the critical component in SQUINT are too simple.SQUINTw/o C model doesn't have much information for algorithmic insight.
- The literature review on relevant work is notably incomplete. Discrete representation learning on transcriptomic data is not new [1] and the comparison with many existing works is missing. A method [2] for spatial imputation and perturbation is also ignored, which makes the claim "To the best of our knowledge, SQUINT is the first model to address this task on SRT data." overstated.
- Claims about batch effect correction are made but not supported by any experimental results.

[1] Li, Y. MetaQ: fast, scalable and accurate metacell inference via single-cell quantization.

[2] Hao, M. *et al.* GeST: Towards Building A Generative Pretrained Transformer for Learning Cellular Spatial Context.

[3] Shang, L. & Zhou, X. Spatially aware dimension reduction for spatial transcriptomics. *Nat Commun* **13**, 7203 (2022).

[4] Tian, T., Zhang, J., Lin, X., Wei, Z. & Hakonarson, H. Dependency-aware deep generative models for multitasking analysis of spatial omics data. *Nat Methods* **21**, 1501–1513 (2024).

**Questions:**

See weakness.

---

> ### Author Response · Authors · 2025-11-28
>
> We thank reviewer yXRW for their constructive comments. Please find below our responses to specific points raised in the review:
>
> **On evaluation of SQUINT**:
> Our methodological contribution is to design discrete tokens for cells using a conditional VQ-VAE model with novel modifications to make it suitable for high resolution, single-cell SRT data.
> - For imputation, SQUINT produces expressions of entire gene expressions of contiguous patches of cells from SRT data. WFM is able to produce this too and hence we compare against it. Our goal is to showcase that the learned tokens can impute entire unseen (and not just reconstruct seen) patches of cells thereby offering usability for generation via downstream models.
> - Our aim with for tasks C and D is to demonstrate that discrete tokens offer inherent biological interpretability (i) for differentiating tumour cells enriched in kidney cancer biomarkers from non-tumour cells and, (ii) for recapitulating known macrophage biology after perturbing states of neighboring T-cells.
>
> **On ablation analysis**:
> We kindly request you to refer to our responses to Reviewers AdAr and Fxmk for additional results.
>
> **On comparisons to previous works**: We thank the reviewer for pointers to these references. Please see our response below to each of the papers in the context of our work:
>
> 1. MetaQ: While this method relies on VQVAE, it has two fundamental differences. One, it is designed for scRNA-seq data modalities where the input is a cell x gene matrix; ours is focused on spatial transcriptomics that includes spatial neighborhood graphs in addition to cell x gene matrices. Two, MetaQ relies on gene expression reconstruction at the cell level for cell-type classification, clustering, and data integration; our method is designed for different tasks such as imputation (not reconstruction, thereby requiring conditioning and masking), tumour stratification, and cell-type perturbation.
> 2. GeST: This method addresses the unseen cell generation task using cell tokens to train a transformer. However, they have not made their code publicly available and therefore we are unable to include this as a baseline in our experiments.
> 3. SpaPCA: This is a dimensionality reduction method for spatial domain discovery, cell-type deconvolution, and trajectory inference. It does not build discrete tokens for cells.
> 4. spVAE: This method uses a variational autoencoder to build continuous embeddings for cells for batch integration, clustering, visualization, identification of spatially variable genes, among others. This is different setting from ours entirely and is thus unsuitable for evaluation as a baseline.
>
> We have updated the manuscript to enhance the discussion on related work to highlight these references and further contextualize our setting and results with past literature. We also include a summary and qualitative comparison in a separate comment above, for your reference.
>
> **On batch correction effect**: We humbly acknowledge this oversight and have corrected the manuscript.

---

### Official Review · Reviewer_DTxe · 2025-10-30

**Soundness:** 3
**Presentation:** 3
**Contribution:** 3
**Rating:** 4
**Confidence:** 3

**Summary:**

This paper presents SQUINT, a spatially-aware discrete representation learning framework for spatial transcriptomics (SRT).
 Unlike continuous embedding approaches such as graph autoencoders or transformers, SQUINT encodes tissues into a finite vocabulary of interpretable discrete codes using a graph neural network encoder combined with vector quantization conditioned on relative spatial distances.
 Each cell is assigned to a shared codebook entry, allowing tissues to be represented as sequences of discrete tokens.
 The learned codes enable gene expression imputation at arbitrary spatial locations and capture recurrent microenvironmental motifs across samples.
 Empirical results on multiple datasets, including 3D skin sections and kidney tumor tissues, show that SQUINT achieves competitive reconstruction accuracy while providing interpretable symbolic representations applicable to downstream tasks such as tumor stratification and perturbation analysis.

**Strengths:**

**Discretization resolution and codebook stability not analyzed:**
 The impact of codebook size, token granularity, and quantization error on tissue representation quality is not systematically evaluated.


**Clear architecture and presentation:**
 The model design—encoder, vector quantization bottleneck, decoder—is well illustrated and easy to follow. Figures and explanations are well aligned with the mathematical description.

**Biological insight:**
 The discrete tokens reveal spatially recurrent structures and microenvironmental organization patterns that correspond to known biological phenomena (e.g., immune infiltration, tumor aggressiveness).

**Weaknesses:**

**Limited novelty relative to existing VQ approaches:**

 The overall framework follows the standard vector quantization autoencoder pipeline (VQ-VAE), with the main innovation being its application to SRT. The methodological contribution may be incremental compared to recent graph-based SRT embedding models.

 **Limited robustness and spatial consistency:**

While SQUINT performs well in 2D spatial imputation, its robustness across 3D tissue sections is limited.
 As acknowledged by the authors, imputation quality drops in the middle of the Z-stack, likely due to misalignment and uneven section spacing.
 This indicates that the current formulation lacks explicit mechanisms to enforce spatial continuity or cross-section alignment, effectively treating 3D data as independent 2D slices.
 Consequently, the model’s generalization and robustness to spatial distortions or imperfect registration remain limited.
 Future work should consider alignment-aware architectures or continuous spatial encoders to achieve true volumetric reconstruction.

**Scalability and computational cost not discussed:**

 Given the large number of cells in modern SRT datasets, it is unclear whether the model scales efficiently to millions of spatial spots or whole-organ datasets.

**Limited cross-platform validation:**

Although SQUINT claims general applicability to spatial transcriptomics data, the experiments are restricted to a small set of platforms (Visium, Xenium, CosMx).
 No evaluation is provided on high-resolution single-cell or subcellular assays such as MERFISH, seqFISH, or Stereo-seq, which differ substantially in spatial density, noise structure, and coordinate geometry.
 Without such validation, it remains unclear whether SQUINT can generalize across measurement technologies or maintain consistent token semantics under varying data modalities.
 This limitation raises questions about the model’s robustness and practical usability for diverse spatial omics datasets.

**Questions:**

None

---

> ### Author Response · Authors · 2025-11-28
>
> We thank reviewer DTxe for their constructive comments. Please find below our responses to specific points raised in the review:
>
> **On novelty relative to existing VQ approaches**:
> - Modeling individual cells as tokens for use by downstream models towards generative tissue-level applications is an important and unsolved challenge in spatial biology.
> - To this end, our methodological contributions are two-fold:
> 	- We design discrete tokens for cells from SRT data that is large in size, with sparse and integer-valued gene expression counts, and frequently unavailable cell-type labels.
> 	- We showcase that the tokens learned by our method have inherent biological interpretability through tasks such as 2D and 3D imputation of entire missing patches, tumour stratification, and cell-type perturbation.
> - This required changes to the architecture (e.g. masking, conditions) and training methodology (e.g. using negative binomial loss, removing supervised cross entropy loss). Existing formulations of VQVAE on graphs such as MoleBERT geared towards predicting labels in molecular graphs or distilling GNNs to MLPs in the case of VQGraph [Yang et al. ICLR 2024] are unsuitable. And existing VAE-based methods for SRT data do not provide discrete tokens with demonstrable interpretability.
> In a separate comment above, we include a qualitative comparison with the previous literature from spatial biology.
>
> [Xia et al, ICLR 2023] Jun Xia, Chengshuai Zhao, Bozhen Hu, Zhangyang Gao, Cheng Tan, Yue Liu, Siyuan Li, and Stan Z, Li, Mole-bert: Rethinking pre-training graph neural networks for molecules
>
> [Yang et al, ICLR 2024] Ling Yang, Ye Tian, Minkai Xu, Zhongyi Liu, Shenda Hong, Wei Qu, Wentao Zhang, Bin CUI, Muhan Zhang, and Jure Leskovec. Vqgraph: Rethinking graph representation space for bridging GNNs and MLPs
>
> **On spatial consistency and future work**:
> We thank the reviewer for this comment and agree. By conditioning on the level within the Z-stack, we have thus far shown that the token embeddings capture the nature of the orientation between slices. Our future goal is to utilize these learned discrete tokens in downstream transformer-style models that can flexibly overcome the misalignment between tissue sections. We have updated the section on future directions to include this.
>
> **On runtime statistics**:
> | Dataset        | Peak Memory   | Runtime               |
> |:---------------|:--------------|:----------------------|
> | Brain (Mouse)  | 4.17 GB       | 2 minutes 52 seconds  |
> | Kidney (Human) | 15.83 GB      | 16 minutes 53 seconds |
> | Skin (Human)   | 16.46 GB      | 6 minutes 1 seconds   |
>
> **On cross-platform validation**:
> Our evaluation is conducted on two high resolution single-cell assays including MERFISH via the Brain (Mouse) dataset and Xenium via the Skin (Human) and Kidney (Human) datasets. Our results show model performance that extends across modalities. We have updated the paragraph on datasets to make this explicit and also updated the table on summary dataset statistics.

---

### Official Review · Reviewer_Fxmk · 2025-11-01

**Soundness:** 2
**Presentation:** 3
**Contribution:** 1
**Rating:** 2
**Confidence:** 5

**Summary:**

The authors present a discrete representation learning framework for ST data that combines GNNs with vector quantization to encode cells as discrete tokens from a learned codebook. The authors claim these tokens enable gene expression imputation at arbitrary spatial locations and demonstrate applications in imputation, tumor stratification, and perturbation analysis. The manuscript in its current state has important conceptual, experimental, and presentation deficiencies that prevent me from recommending acceptance. The core concerns are: weak baseline comparisons, insufficient ablations and controls for important design choices, potential data/experimental leakage, over-claiming interpretability/clinical relevance from tiny cohorts, and missing reproducibility details.

**Strengths:**

- The paper demonstrates the utility of learned discrete tokens across multiple biologically relevant applications (2D/3D imputation, tumor stratification, perturbation analysis), showing that the representations capture meaningful information beyond simple reconstruction.
- Table 1 shows SQUINT achieves substantially better reconstruction metrics than the baseline across multiple datasets and species.

**Weaknesses:**

- Authors claim the method enable generation of expression profiles. However the evaluation is limited to reconstruction metrics (MSE, SSIM, Pearson correlation). The authors should present examples of samples from the generative model, assessments of sample diversity, evaluations of biological validity beyond held-out reconstruction and comparisons to baseline generative models.
- The authors fail to cite or compare against the relevant prior work. The paper claims to be "the first spatially-aware discrete tokenizer tailored to SRT". These claims are demonstrably false e.g., Yarlagadda et. al. [1] which presents a similar approach using VQ-VAE for ST data. This is oversight undermines claims of novelty.
- The only generative baseline is WFM, which requires cell-type annotations & cannot perform location-specific imputation. The authors exclude other GNN-based ST methods like SpaGCN, STAGATE, GraphST, recent transformer approaches like CellPLM and scGPT-spatial, NMF-/ topic-modeling based approaches, or scVI-like VAEs adapted for spatial data.
- How important are the different components like masking and FiLM conditioning? An ablation study assessing their impact is lacking. The "custom masking strategy" (Eq. 4) is just replacing masked cells with a learnable vector plus noise.
- The generative model formulation (Section 4.1) is unnecessarily complex for what amounts to a conditional VQ-VAE
- Why is GraphSAGE with mean pooling used here? The single-layer GNN seems simplistic. Fig. 8 shows GIN performs comparably—was this actually optimized or just chosen arbitrarily?
- The MSSIM + NB loss combination (Eq. 5) lacks justification. Why MSSIM for sparse gene expression data? The λ weights appear hand-tuned without principled selection.
- The imputation experiments define patches within one section as imputation sites while training on other cells from that section as well as remaining sections. That phrasing suggests the training set may include spatially adjacent cells to the held-out patches, risking trivial interpolation rather than true out-of-region generalization. The authors must precisely define the train/test partitioning (are held-out patches contiguous? how far from training cells?), and ideally evaluate far-away masking splits (e.g., hold out entire anatomical regions or whole sections) to test true generalization. Without this, the strong numerical gains could reflect spatial proximity rather than real generative power.

[1]. Yarlagadda, D. V. K., Massagué, J., & Leslie, C. (2023). Discrete representation learning for modeling imaging-based spatial transcriptomics data. In Proceedings of the IEEE/CVF International Conference on Computer Vision (pp. 3846-3855).

**Questions:**

- How is the codebook initialized? The authors state using a codebook size of 5000 with multiple heads in some experiments (and claim it “avoids collapse”), but in the 3D skin experiment they use a codebook size of 50 (and 200 latent dims) — no explanation for the dramatic difference or guidance on choosing K. There is no systematic ablation of codebook size, head count, code utilization statistics, or metrics of codebook collapse.
- What is the masking schedule (0.2 to 0.6 annealing)?
- How stable is training with the straight-through estimator? What is the variance across random seeds?

---

> ### Author Response · Authors · 2025-11-28
>
> We thank reviewer Fxmk for their constructive comments. Please find below our responses to specific points raised in the review:
>
> **On MSE, SSIM, Pearson reconstruction metrics**:
> Our experiments do not validate against reconstruction metrics such as MSE or SSIM. We use mean Pearson correlation to measure accuracy to expression profiles of cells at the specified imputation sites and Energy and Mean Maximum Discrepancy metrics to measure the quality of the generated cell samples (similar to the setting from WFM). Through tasks C, and D, we showcase the biological validity of the learned discrete tokens for tumour stratification and cell perturbations.
>
> **On previous work by Yarlagadda, et al.**:
> We thank the reviewer for the pointer to this work. We note that there are three key differences that make their setting materially different to ours and therefore unsuitable as a baseline:
> - Their data is high-resolution MERFISH *images* and requires vision-based preprocessing to approximately identify and assign transcripts to cells, our setting is based on high-resolution single-cell transcriptomic data directly obtained from sequencing technologies such as Xenium and MERFISH.
> - The discrete tokens learned in their setting model blocks of pixels for spatially-variable genes and subcellular structures, our discrete tokens model cells and their surrounding microenvironments
> - Their setting evaluates the reconstruction quality of images generated by their model, our setting defines imputing gene expression of entire patches of cells while also showcasing the interpretability of the learned codebook through tumor stratification and cell-type perturbation.
>
> **On previous related work such as SpaGCN, STAGATE, etc.**:
> We kindly refer the reviewer to our separate comment above wherein we include a consolidated qualitative comparison with previous literature from spatial biology, including the ones mentioned in this review, to further contextualize our contributions.
>
> We humbly accept the reviewer’s comment on our broad statement on the first discrete tokenizer and update our manuscript to clarify this language wherever applicable.
>
> **On learnable masking**:
> Masking is crucial because our setting is imputation and not reconstruction. Since gene expression tends to be sparse, we observed that a learnable mask token offers better performance than zero masking since SRT data tends to be sparse with zero expression for many genes (line 172). Zero masking causes the model to predict zero-expression for 7.2%, 3.1%, and 9.9% of cells from the imputation patches in the Brain (Mouse), Kidney (Human), and Skin (Human) datasets, respectively. This is a key concern because these datasets have been preprocessed to remove zero-expression cells. We provide ablation results for MMD scores (lower is better) averaged across three seeds below and update the manuscript (Figure 8, Appendix C.3).
>
> | Mask      | Brain (Mouse)       | Kidney (Human)      | Skin (Human)        |
> |:----------|:--------------------|:--------------------|:--------------------|
> | Learnable | 0.0395 $\pm$ 0.0015 | 0.0194 $\pm$ 0.0015 | 0.0927 $\pm$ 0.0054 |
> | Zero      | 0.1034 $\pm$ 0.0047 | 0.0409 $\pm$ 0.0023 | 0.1160 $\pm$ 0.0003 |
>
> **On model formulation and MSSIM and NB losses**:
> We detail the inference and generative pathways to account for the masking strategy alongside conditions, with a view to provide completeness (Section 4.1). For improved readability, we update section to explicitly state that this is a conditional VQ-VAE.
>
> Our training objective does not include an MSSIM loss (MSSIM loss is used by Yarlaggada, et al.). The generative pathway described in Section 4.1 (and its derivation included in Appendix A.2 for completeness) uses negative binomial loss for attributes since these represent raw counts and binary cross entropy loss for adjacency which is binary (undirected spatial neighborhood graph).
>
> **On imputation sites**:
> We observed that holding out three patches within one section as imputation sites (line 308) is more challenging than imputing expression profiles for cellular microenvironments randomly selected from across the entire tissue slice. The large performance gap may be attributed to the fact that in the random setting, the model has access to expression of the microenvironment around the imputation site (i.e. cell) whereas this is not the case in the contiguous patch setting. We report results below (averaged over 3 seeds) and update the manuscript (Table 3, Appendix C.4).
>
> | Site      | Brain (Mouse)       | Kidney (Human)      | Skin (Human)        |
> |:----------|:--------------------|:--------------------|:--------------------|
> | 3 Patches | 0.0395 $\pm$ 0.0015 | 0.0194 $\pm$ 0.0015 | 0.0927 $\pm$ 0.0054 |
> | Random    | 0.0139 $\pm$ 0.0009 | 0.0082 $\pm$ 0.0001 | 0.0129 $\pm$ 0.0005 |

---

> > ### Author Response · Authors · 2025-12-01
> >
> > (continued...)
> >
> > **On additional questions**:
> > - Codebooks: The codebook is initialized using a uniform distribution over the number of codebooks, codebook size, and dimensionality followed by L2 normalization. Codebook collapse was evaluated by measuring the mean pairwise similarity of the learned codes (0.19 for Brain (Mouse), 0.22 for Skin (Human), and 0.13 for Kidney (Human)). Across our experiments, we observed >95% codebook utilization during training demonstrating high coverage/utility across codes. We conducted the 3D imputation experiment with a small codebook size (50) to demonstrate the effect of code assignment conditioned on the level of the tissue section in the Z-stack and evaluate token co-occurrence.
> > - Masking schedule: The masking schedule increases linearly from 0.2 to 0.6 across epochs.
> > - Variance: We update Table 2 with variance across seeds.

---

### Official Review · Reviewer_AdAr · 2025-11-01

**Soundness:** 2
**Presentation:** 3
**Contribution:** 2
**Rating:** 4
**Confidence:** 4

**Summary:**

This paper introduces SQUINT, a novel framework designed to learn discrete representations from spatially-resolved transcriptomics (SRT) data. The primary goal is to capture tissue-level organization and microenvironmental details within a finite, interpretable vocabulary of discrete codes. The method combines a Graph Neural Network (GNN) encoder with vector quantization (VQ), conditioned on relative spatial distances and trained using a masking strategy. Cells are thus represented by token assignments, enabling whole tissues to be modeled as sequences. The authors validate their approach on several downstream tasks, including 2D and 3D gene expression imputation, tumour stratification, and in silico perturbation analysis, claiming SQUINT provides interpretable codes and outperforms existing generative methods.

**Strengths:**

- The paper tackles the important and challenging problem of learning compact, interpretable, and reusable representations for complex SRT data, which is a significant bottleneck in the field.
- The proposed method of combining a GNN with vector quantization to create a "tissue tokenizer" is a novel and interesting approach.
- The authors demonstrate the utility of the learned discrete codes across a diverse range of downstream applications (imputation, stratification, perturbation), showing the potential versatility of the framework.

**Weaknesses:**

1. **Overstated Novelty and Missing Baselines:** The authors claim that SQUINT is "the first model to address" the task of imputing gene expression at specific spatial locations. This is factually incorrect. For example, scGPT-spatial already supports this task and explicitly includes it as a pre-training objective. A direct performance comparison with this highly relevant baseline is missing, which makes it difficult to assess the actual performance and contribution of SQUINT for spatial imputation.
2. **Unaddressed Scalability Concerns:** The paper does not sufficiently address the scalability of the GNN-based approach. The largest dataset mentioned (Sec 5.2) contains 280K cells, which is modest by modern SRT standards. It is unclear how SQUINT would perform in terms of memory and runtime on datasets with millions of cells. The authors do not report detailed computational overhead nor do they discuss potential GNN-specific issues like over-smoothing, which can be a significant problem in large, densely-connected graphs.
3. **Insufficient Justification for Architectural Choices:** The justification for key architectural choices is lacking. The paper asserts that a two-layer MLP is sufficient to handle data sparsity, but this claim is not substantiated with ablation studies or other evidence. It is unclear how this conclusion was reached or why this specific design is optimal.
4. **Lack of Detailed Dataset Information:** The paper omits a clear, consolidated summary of the datasets. For reproducibility, it is essential to provide specific details for *each* dataset, including the precise cell counts, the sequencing technology used (e.g., Xenium, Visium), and the number of genes measured.

**Questions:**

1. Could the authors please clarify the claim of novelty regarding spatial imputation, given that models like scGPT-spatial already perform this task? More importantly, could you provide a direct performance comparison against scGPT-spatial for the 2D expression imputation task?
2. Regarding scalability:
   - What are the specific runtime and peak memory usage for training SQUINT on the 280K-cell dataset (Task B)? How do you anticipate these resources scaling to datasets with >1 million cells?
   - Have you investigated the potential for GNN over-smoothing in your framework, particularly as the graph size and neighborhood depth increase?
3. Could the authors provide an ablation study or further justification for the claim that a two-layer MLP is sufficient for handling data sparsity?
4. Could you please add a table summarizing the key statistics for *each* dataset used in the experiments (e.g., specific cell count, sequencing method, gene count, data source)?

---

> ### Author Response · Authors · 2025-11-28
>
> We thank reviewer AdAr for their constructive comments. Please find below our responses to specific points raised in the review:
>
> **On scGPT-spatial and related work**:
> We discuss scGPT-spatial in our section on Related Work (line 94). Their gene expression imputation task is to generate expression of a subset of missing genes from Xenium slides using paired Visium data (lung tissue samples) where each gene is tokenized by a unique integer identifier. Our expression imputation task is to impute the entire expression profile of unseen patches by tokenizing cells without requiring any reference or paired data. This is why we believe scGPT-spatial is unsuitable as a baseline for our experiments. We humbly accept the comment on our broad statement of “first model” and update our manuscript to qualify the language to more accurately reflect our contributions.
> We kindly refer the reviewer to our separate comment above wherein we include a consolidated qualitative comparison with previous literature from spatial biology, including the ones mentioned in this review, to further contextualize our contributions.
>
> **On oversmoothing**:
> The issue of oversmoothing has been typically observed for deep GNN models where the number of layers is larger than the problem radius [Alon, et al, ICLR 2021]. SQUINT uses a single GNN layer and the neighborhood spatial graph is constructed from 8 nearest neighbors. The average diameter and average degree of the spatial neighborhood graph of tissue slices in our datasets are large (>50 and >20 respectively). We include this as a discussion in the appendix.
>
> [Alon, et al, ICLR 2021] On the bottleneck of graph neural networks and its practical implications.
>
> **On MLP layers**:
> In our experiments, we observed that a 2-layer MLP offers improved imputation quality scores over a 1-layer MLP as well as over not having an MLP at all. We report MMD scores (lower is better) averaged across three seeds below and update the manuscript (Figure 7, Appendix C.2).
>
> |   MLP Layers | Brain (Mouse)       | Kidney (Human)      | Skin (Human)        |
> |-------------:|:--------------------|:--------------------|:--------------------|
> |            0 | 0.1365 $\pm$ 0.0065 | 0.0265 $\pm$ 0.0035 | 0.1445 $\pm$ 0.0184 |
> |            1 | 0.0935 $\pm$ 0.0253 | 0.0248 $\pm$ 0.0027 | 0.1346 $\pm$ 0.0104 |
> |            2 | 0.0395 $\pm$ 0.0015 | 0.0194 $\pm$ 0.0015 | 0.0927 $\pm$ 0.0054 |
>
> **On missing dataset information**:
> Due to space limitations, we included dataset-level summary statistics in Appendix Section B.1 and referenced this in the main text (line 306). For improved readability, we update the manuscript to explicitly highlight that the Skin (Human) and Kidney (Human) are Xenium datasets and Brain (Mouse) is assayed using MERFISH technology.

---

### Author Response · Authors · 2025-11-28

We sincerely thank all the reviewers for engaging with our submission in detail. Please find below a summary clarifying our contributions and contextualizing previous literature mentioned across separate reviews.

SQUINT builds discrete and interpretable tokens for cells from high-resolution single-cell spatial transcriptomics data from MERFISH and Xenium assays using spatial neighborhood graphs and cell-by-gene matrices. We showcase generative capabilities of these tokens through cell imputation of unseen regions. We achieve this through a graph-based VQVAE architecture with conditioning and masking during training. We demonstrate their interpretability through tumour stratification and cell-type perturbation.

| Model                   | Methodology              | Data Modalities                            | Tokenization          | Tasks                               |
| ----------------------- | ------------------------ | ------------------------------------------ | --------------------- | ----------------------------------- |
| SpaPCA                  | Dimensionality reduction | Spatial transcriptomics                    | Not applicable        | Niche identification, deconvolution |
| spVAE, SpaGCN, STAGATE… | VAE                      | Spatial transcriptomics                    | Continuous, gene-wise | Cell/niche annotation, Clustering   |
| Yarlagadda et al.       | VQVAE                    | Images                                     | Discrete, cell-wise   | Reconstruction                      |
| MetaQ                   | VQVAE                    | scRNA-seq                                  | Discrete, cell-wise   | Reconstruction, clustering          |
| GeST                    | Transformer                    | Spatial transcriptomics                    | Discrete, cell-wise   | Unseen cell generation              |
| scGPT-spatial           | Transformer              | Paired spatial and reference sequence data | Discrete, gene-wise   | Missing gene imputation, clustering |
| CellPLM                 | Transformer              | Paired spatial and reference sequence data | Discrete, cell-wise   | Missing gene imputation, annotation |
| SQUINT                  | (Conditional) VQVAE      | Spatial transcriptomics                    | Discrete, cell-wise   | Unseen cell generation              |

For GeST, we note that we were unable to find public code to benchmark against.

---

### Meta-Review · Area_Chair_tGVv · 2026-01-07

**Summary:**

Reviewers agree the problem is important and the idea is potentially useful, but the overall reception is negative: two reviewers give strong reject (2) with high confidence, and two give borderline (4). The main drivers are (i) novelty concerns relative to prior VQ/discretization and graph/transformer SRT models, (ii) insufficient and uneven baselines/quantitative comparisons (especially beyond 2D imputation), and (iii) missing or under-justified design/robustness/scalability analyses (notably for large graphs and 3D misalignment).

**Reviewer Concerns:**

### Addressed (partially)

- Softened/clarified “first” novelty claims; expanded related-work positioning.

- Added/clarified ablations (masking choice, imputation-site protocol), reported additional distributional metrics, and provided some implementation/runtime/memory details and variance notes.

- Clarified codebook initialization/utilization and corrected an unsupported batch-correction claim.

### Still outstanding

- Baseline gap remains the central issue: reviewers asked for stronger, task-appropriate quantitative comparisons (multiple suggested SRT baselines and tokenization/generative alternatives). The rebuttal argues some are mismatched or unavailable, but does not fully resolve the lack of head-to-head evidence.

- Evidence for “generation” and biological validity beyond the showcased settings remains limited; interpretability claims are still viewed as under-supported by systematic analysis.

- Robustness (misalignment/section spacing) and scalability (toward very large tissues / million-cell graphs) are still not convincingly addressed; several architectural choices remain only lightly justified.

**Reviewer Scores:**

R1 (AdAr): 4 -> 4: clarifications/added analyses help, but baseline/scalability gaps remain

R2 (DTxe): 4 -> 4: runtime/cross-platform notes help; novelty/3D robustness still concerns

R3 (Fxmk): 2 -> 2/3: rebuttal addresses some misunderstandings and adds ablations, but major missing comparisons/validation remain

R4 (yXRW): 2 -> 2/3: evaluation depth and baselines remain the blocker

---

### Decision · Program_Chairs · 2026-01-26

Reject